# RLLTE: Long-Term Evolution Project of Reinforcement Learning

## Abstract

We present RLLTE: a long-term evolution, extremely modular, and open-source framework for reinforcement learning (RL) research and application. Beyond delivering top-notch algorithm implementations, RLLTE also serves as a toolkit for developing algorithms. More specifically, RLLTE decouples the RL algorithms completely from the exploitation-exploration perspective, providing a large number of components to accelerate algorithm development and evolution. In particular, RLLTE is the first RL framework to build a complete and luxuriant ecosystem, which includes model training, evaluation, deployment, benchmark hub, and large language model (LLM)-empowered copilot. RLLTE is expected to set standards for RL engineering practice and be highly stimulative for industry and academia.

## 1 Introduction

Reinforcement learning (RL) has emerged as a highly significant research topic, garnering considerable attention due to its remarkable achievements in diverse fields, including smart manufacturing and autonomous driving (Mnih et al., 2015; Duan et al., 2016; Schulman et al., 2017; Haarnoja et al., 2018; Yarats et al., 2021). However, the efficient and reliable engineering implementation of RL algorithms remains a long-standing challenge. These algorithms often possess sophisticated structures, where minor code variations can substantially influence their practical performance. Academia requires a stable baseline for algorithm comparison, while the industry seeks convenient interfaces for swift application development (Raffin et al., 2021). However, the design and maintenance of an RL library prove costly, involving substantial computing resources, making it prohibitive for most research teams.

To tackle this problem, several open-source projects were proposed to offer reference implementations of popular RL algorithms (Liang et al., 2018; D'Eramo et al., 2021; Fujita et al., 2021; Raffin et al., 2021; Huang et al., 2022). For instance, Raffin et al. (2021) developed a stable-baselines3 (SB3) framework, which encompasses seven model-free deep RL algorithms, including proximal policy optimization (PPO) (Schulman et al., 2017) and asynchronous actor-critic (A2C) (Mnih et al., 2016). SB3 prioritizes stability and reliability, and rigorous code testing has been conducted to minimize implementation errors and ensure the reproducibility of results. Weng et al. (2022a) introduced Tianshou, a highly modularized library emphasizing flexibility and training process standardization. Tianshou also provides a unified interface for various algorithms, such as offline and imitation learning. In contrast, Huang et al. (2022) introduced CleanRL, which focuses on single-file implementations to facilitate algorithm comprehension, new features prototyping, experiment analysis, and scalability.

Despite their achievements, most of the existing benchmarks have not established a long-term evolution plan and have proven to be short-lived. Firstly, the consistent complexity of RL algorithms naturally results in distinct coding styles, posing significant obstacles to open-source collaborations. Complete algorithm decoupling and modularization have yet to be well achieved, making maintenance challenging and limiting extensibility. Secondly, these projects are deficient in establishing a comprehensive application ecosystem. They primarily concentrate on model training, disregarding vital aspects like model evaluation and deployment. Furthermore, they frequently lack exhaustive benchmark testing data, including essential elements like learning curves and trained models. This deficiency makes replicating algorithms demanding in terms of computational resources.

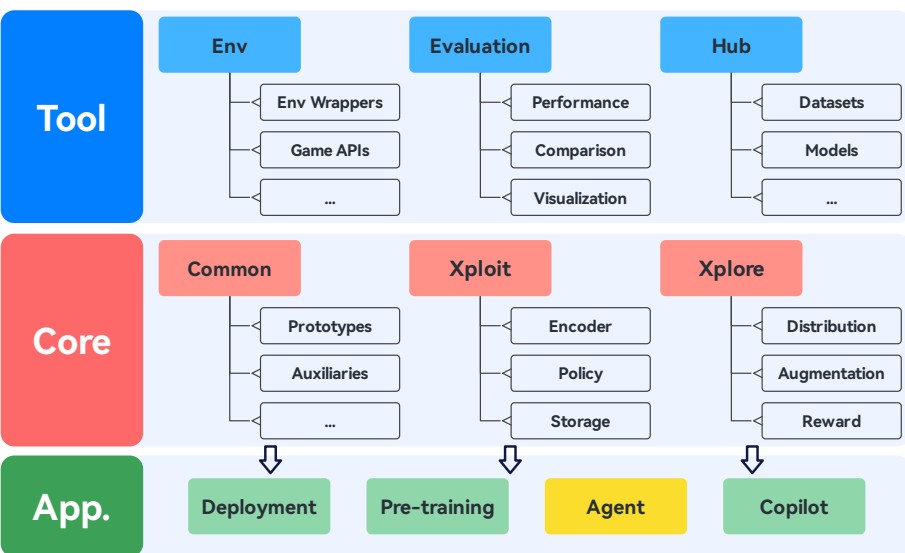

Figure 1: Overview of the architecture of RLLTE.

Inspired by the discussions above, we propose **RLLTE**, a long-term evolution, extremely modular, and open-source framework of RL. We summarize the highlighted features of RLLTE as follows:

- **Module-oriented.** RLLTE decouples RL algorithms from the *exploitation-exploration* perspective and breaks them down into minimal primitives, such as *encoder* for feature extraction and *storage* for archiving and sampling experiences. RLLTE offers a rich selection of modules for each primitive, enabling developers to utilize them as building blocks for constructing algorithms. As a result, the focus of RLLTE shifts from specific algorithms to providing more handy modules like PyTorch. In particular, each module in RLLTE is customizable and plug-and-play, empowering users to develop their own modules. This decoupling process also contributes to advancements in interpretability research, allowing for a more in-depth exploration of RL algorithms.

- **Long-term evolution.** RLLTE is a long-term evolution project, continually involving advanced algorithms and tools in RL. RLLTE will be updated based on the following tenet: (i) generality; (ii) improvements in generalization ability and sample efficiency; (iii) excellent performance on recognized benchmarks; (iv) promising tools for RL. Therefore, this project can uphold the right volume and high quality resources, thereby inspiring more subsequent projects.

- **Data augmentation.** Recent approaches have introduced data augmentation techniques at the *observation* and *reward* levels to improve the sample efficiency and generalization ability of RL agents, which are cost-effective and highly efficient. In line with this trend, RLLTE incorporates built-in support for data augmentation operations and offers a wide range of observation augmentation modules and intrinsic reward modules.

- **Abundant ecosystem.** RLLTE considers the needs of both academia and industry and develops an abundant project ecosystem. For instance, RLLTE designed an evaluation toolkit to provide statistical and reliable metrics for assessing RL algorithms. Additionally, the deployment toolkit enables the seamless execution of models on various inference devices. In particular, RLLTE attempts to introduce the large language model (LLM) to build an intelligent copilot for RL research and applications.

- **Comprehensive benchmark data.** Existing RL projects typically conduct testing on a limited number of benchmarks and often lack comprehensive training data, including learning

curves and test scores. While this limitation is understandable, given the resource-intensive nature of RL training, it hampers the advancement of subsequent research. To address this issue, RLLTE has established a data hub utilizing the Hugging Face platform. This data hub provides extensive testing data for the included algorithms on widely recognized benchmarks. By offering complete and accessible testing data, RLLTE will facilitate and accelerate future research endeavors in RL.

- **Multi-hardware support.** RLLTE has been thoughtfully designed to accommodate diverse computing hardware configurations, including graphic processing units (GPUs) and neural network processing units (NPUs), in response to the escalating global demand for computing power. This flexibility enables RLLTE to support various computing resources, ensuring optimal trade-off of performance and scalability for RL applications.

## 2  ARCHITECTURE

Figure 1 illustrates the overall architecture of RLLTE, which contains the core layer, application layer, and tool layer. The following sections will detail the design concepts and usage of the three layers.

### 2.1  CORE LAYER

In the core layer, we decouple an RL algorithm from the *exploitation-exploration* perspective and break them down into minimal primitives. Figure 2 illustrates a typical forward and update workflow of RL training. At each time step, an encoder first processes the observation to extract features. Then, the features are passed to a policy module to generate actions. Finally, the transition will be inserted into the storage, and the agent will sample from the storage to perform the policy update. In particular, we can use data augmentation techniques such as observation augmentation and intrinsic reward shaping to improve the sample efficiency and generalization ability.

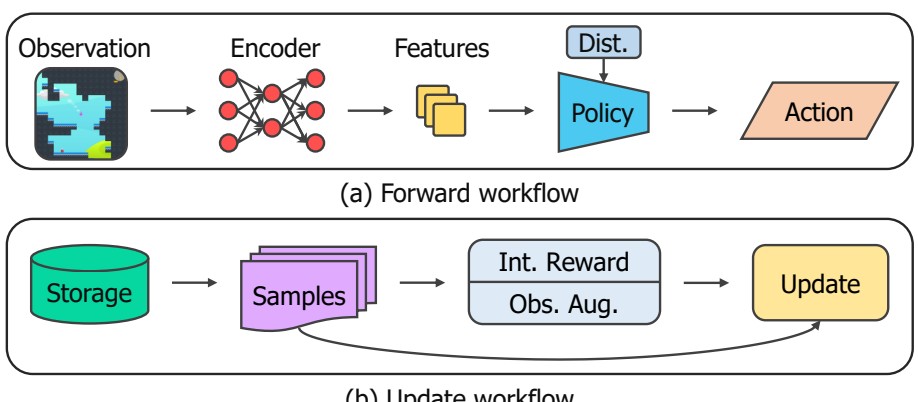

Figure 2: Forward and update workflow of an RL algorithm. **Aug.**: Augmentation. **Dist.**: Distribution for sampling actions. **Int.**: Intrinsic. **Obs.**: Observation.

We categorize these fundamental components into two parts: `xploit` and `xplore`, and Table 1 illustrates their architectures. The modules within the `xploit` component primarily focus on exploiting the current collected experiences. For instance, the `storage` module defines the methods for storing and sampling experiences, while the `policy` module is updated based on the sampled data. In contrast, modules in `xplore` focus on exploring unknown domains. When `policy` is stochastic, `distribution` specifies the methods for sampling actions from the action space. In the case of a deterministic `policy`, the `distribution` module introduces noise to the current action to enhance the exploration of the action space. The `augmentation` and `reward` modules contribute to exploring the state and action space by augmenting observations and providing additional intrinsic rewards, respectively. Each submodule in Table 1 is accompanied by many pre-defined components, which are listed in Appendix A.

Table 1: Six primitives in RLLTE. Note that the action noise is implemented via a distribution manner to keep unification in RLLTE.

| Module | Submodule | Remark |
|---|---|---|
| rllte.xploit | policy
encoder
storage | Policies for interaction and learning.
Encoders for feature extraction.
Storages for collected experiences. |
| rllte.xplore | distribution
augmentation
reward | Distributions for sampling actions.
Observation augmentation modules.
Intrinsic reward modules. |

## 2.2 APPLICATION LAYER

Equipped with modules of the core layer, we can efficiently develop RL algorithms and applications with simple steps, and Table 2 illustrates the architecture of the application layer. See all the corresponding code examples in Appendix C.

Table 2: Architecture of the application layer in RLLTE.

| Module | Remark |
|---|---|
| rllte.agent | Top-notch implementations of highly-recognized RL algorithms, in which convenient interfaces are designed to realize fast application construction. In particular, the module-oriented design allows developers to replace settled modules of implemented algorithms to make performance comparisons and algorithm improvements. |
| Pre-training | Since RLLTE is designed to support intrinsic reward modules natively, developers can conveniently realize pre-training. The pre-trained weights will be saved automatically after training, and it suffices to perform fine-tuning by loading the weights in the .train() function. |
| Deployment | A toolkit that helps developers run their RL models on inference devices, which consistently have lower computational power. RLLTE currently supports two inference frameworks: NVIDIA TensorRT and HUAWEI CANN. RLLTE provides a fast API for model transformation and inference, and developers can invoke it directly with their models. |
| Copilot | A promising attempt to introduce the LLM into an RL framework. The copilot can help users reduce the time required for learning frameworks and assist in the design and development of RL applications. We are developing more advanced features to it, including RL-oriented code completion and training control. |

### 2.2.1 FAST ALGORITHM CONSTRUCTION

Developers only need three steps to implement an RL algorithm with RLLTE: (i) select an algorithm prototype; (ii) select desired modules; (iii) define an update function. Currently, RLLTE provides three algorithm prototypes: `OnPolicyAgent`, `OffPolicyAgent`, and `DistributedAgent`. Figure 3 demonstrates how to write an A2C agent for discrete control tasks with RLLTE:

As shown in this example, developers can effortlessly choose the desired modules and create an `.update()` function to implement a new algorithm. At present, the framework includes a collection of 13 algorithms, such as data-regularized actor-critic (DrAC) (Raileanu et al., 2021) and data regularized Q-v2 (DrQ-v2), and the detailed introduction can be found in Appendix B.

```python
from rllte.common.prototype import OnPolicyAgent
from rllte.xploit.encoder import MnihCnnEncoder
from rllte.xploit.policy import OnPolicySharedActorCritic
from rllte.xploit.storage import VanillaRolloutStorage
from rllte.xplore.distribution import Categorical

class A2C(OnPolicyAgent):
    def __init__(self, ...) -> None:
        super().__init__(...)
        # create essential modules
        encoder = MnihCnnEncoder(...)
        policy = OnPolicySharedActorCritic(...)
        storage = VanillaRolloutStorage(...)
        dist = Categorical()
        # set all the modules
        self.set(encoder=encoder, policy=policy,
                 storage=storage, distribution=dist)

    def update(self) -> Dict[str, float]:
        batch = self.storage.sample()
        # A2C update rule
        ...
```

```python
# import `env` and `agent`
from rllte.env import make_dmc_env
from rllte.agent import DrQv2

if __name__ == "__main__":
    device = "cuda:0"
    # create env, `eval_env` is optional
    env = make_dmc_env(env_id="cartpole_balance",
                       device=device)
    eval_env = make_dmc_env(env_id="cartpole_balance",
                            device=device)
    # create agent
    agent = DrQv2(env=env,
                  eval_env=eval_env,
                  device=device,
                  tag="drqv2_dmc_pixel")
    # start training
    agent.train(num_train_steps=500000,
                log_interval=1000)
```

Figure 3: **Left**: Implement A2C algorithm with dozens of lines of code, and the complete code example can be found in Appendix C.1. **Right**: Simple interface to invoke implemented RL algorithms.

### 2.2.2  MODULE REPLACEMENT

For an implemented algorithm, developers can replace its settled modules using the `.set()` method to realize performance comparisons and algorithm improvements. Moreover, developers can utilize custom modules as long as they inherit from the base class, as demonstrated in the code example in Appendix C.2. By decoupling these elements, RLLTE also empowers developers to construct prototypes and perform quantitative analysis of algorithm performance swiftly.

### 2.2.3  COPILOT

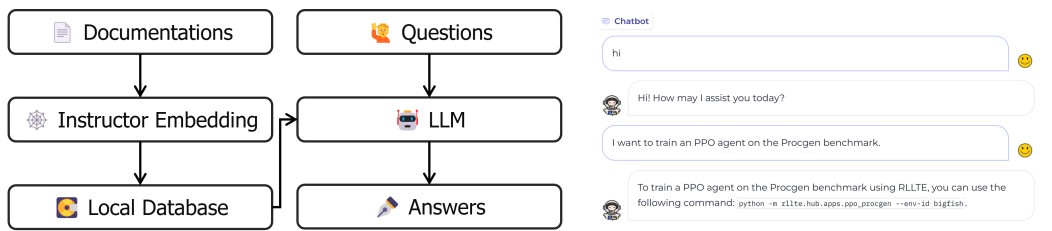

Figure 4: **Left**: The workflow of the copilot. **Right**: A conversation example of training an PPO agent using RLLTE.

**Copilot** is the first attempt to integrate an LLM into an RL framework, which aims to help developers reduce the learning cost and facilitate application construction. We follow the design of (Toro, 2023) that interacts privately with documents using the power of GPT, and Figure 4 illustrates its architecture. The source documents are first ingested by an instructor embedding tool to create a local vector database. After that, a local LLM is used to understand questions and create answers based on the database. In practice, we utilize Vicuna-7B (Chiang et al., 2023) as the base model and build the database using various corpora, including API documentation, tutorials, and RL references. The powerful understanding ability of the LLM model enables the copilot to accurately answer questions about the use of the framework and any other questions of RL. Moreover, no additional training is required, and users are free to replace the base model according to their computing power. In future work, we will further enrich the corpus and add the code completion function to build a more intelligent copilot for RL.

### 2.3  TOOL LAYER

The tool layer provides practical toolkits for task design, model evaluation, and benchmark data. `rllte.env` allows users to design task environments following the natural Gymnasium pattern with-

Table 3: Architecture of the tool layer in RLLTE. Code example for each toolkit can be found in Appendix D.

| Toolkit | Remark |
| --- | --- |
| `rllte.env` | Provides a large number of packaged environments (e.g., Atari games) for fast invocation. RLLTE is designed to natively support Gymnasium (Towers et al., 2023), which is a maintained fork of the Gym library of OpenAI (Brockman et al., 2016). Moreover, developers are allowed to use their custom environments with built-in wrappers in RLLTE. |
| `rllte.evaluation` | Provides reasonable and reliable metrics for algorithm evaluation following (Agarwal et al., 2021). `Performance` module for evaluating a single algorithm. `Comparison` module for comparing multiple algorithms. `Visualization` for visualizing computed metrics. |
| `rllte.hub` | Provides a large number of reusable datasets (`.datasets`) and trained models (`.models`) of supported RL algorithms. Developers can also reproduce the training process via the pre-defined RL applications (`.applications`). |

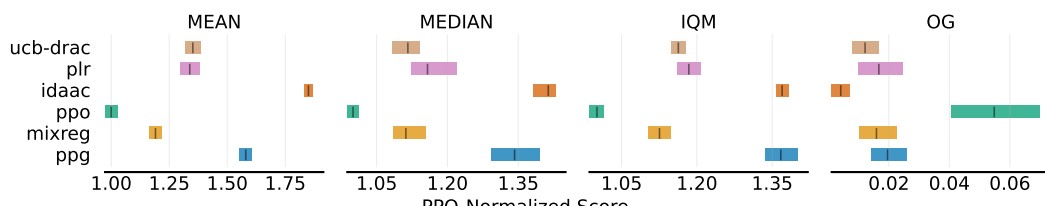

(a) Aggregate metrics with 95% confidence intervals (CIs). **IQM**: Interquartile mean. **OG**: Optimality gap.

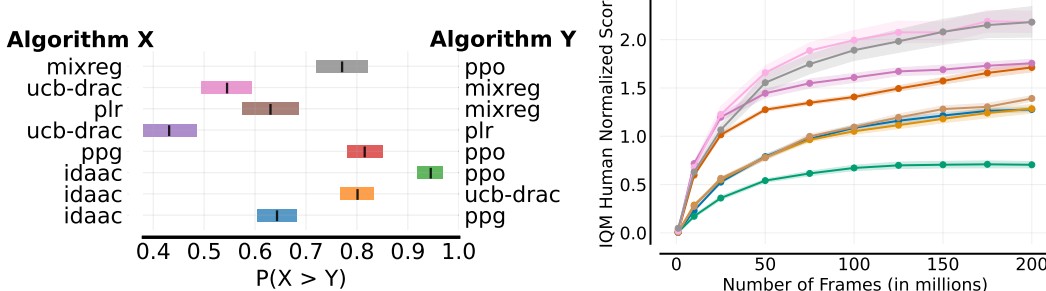

(b) **Left**: Each row shows the probability of improvement, with 95% bootstrap CIs, that the algorithm $X$ on the left outperforms algorithm $Y$ on the right. **Right**: Sample-efficiency of agents as a function of number of frames measured via IQM human-normalized scores.

Figure 5: Performance metrics computed and visualized by `rllte.evaluation`, and the code example can be found in Appendix D.2.

out additional effort. All the environments in RLLTE are set to be vectorized to guarantee sample efficiency, and many different observation and action spaces (e.g., box, discrete, multi-binary, etc.) are supported. In particular, users can also use EnvPool (Weng et al., 2022b) to realize ultra-fast operational acceleration. See code example in Appendix D.1.

Beyond providing efficient task design and training interfaces, RLLTE further investigates the model evaluation problem in RL and develops a simple evaluation toolkit. RLLTE reconstructs and improves the code of (Agarwal et al., 2021) to realize a more convenient and efficient interface. Figure 5 illustrates several metrics computed and visualized by the toolkit.

Finally, `rllte.hub` can accelerate academic research by providing practically available benchmark data, including training data and trained models. This toolkit will save much time and computational resources for researchers, and the code example can be found in Appendix D.3. RLLTE is the first open-source RL project that aims to build a complete ecosystem. Developers can perform task design, model training, model evaluation, and model deployment within one framework. As a result, RLLTE is highly stimulative for both industry and academia.

## 3  PROJECT EVOLUTION

As a long-term evolution project, RLLTE is expected to consistently provide high-quality and timely engineering standards and development components for RL. To that end, RLLTE sets the following tenet for updating new features:

- Generality is the most important;
- Improvements in sample efficiency or generalization ability;
- Excellent performance on recognized benchmarks;
- Promising tools for RL.

Firstly, RLLTE only accepts general algorithms that can be applied in many distinct scenarios and tasks. For example, PPO is a general RL algorithm that can solve tasks with arbitrary action spaces, and random network distillation (RND) (Burda et al., 2019) is a general intrinsic reward module that can be combined with arbitrary RL agents. This rule can effectively control the volume of the project while ensuring its adaptability to a wide range of requirements. Moreover, generality exemplifies the potential for future enhancements (e.g., the various variants of PPO), which can also reduce the difficulty of open-source collaboration and maintain community vitality. Furthermore, the algorithm is expected to improve sample efficiency or generalization ability (e.g., better intrinsic reward shaping approaches), two long-standing and critical problems in RL. Accordingly, the algorithm must be evaluated on multiple recognized benchmarks like Atari (Bellemare et al., 2013) and Procgen games (Cobbe et al., 2020) to guarantee practical performance across tasks. In particular, RLLTE also accepts various promising tools (e.g., operational efficiency optimization, model evaluation, and deployment) to maintain a comprehensive ecosystem. In summary, RLLTE will keep evolving to adapt to changing needs and produce a positive impact on the RL community.

Table 4: Architecture comparison with existing projects. **Modularized**: The project adopts a modular design with reusable components. **Parallel**: The project supports parallel learning. **Decoupling**: The project supports algorithm decoupling and module replacement. **Backend**: Which machine learning framework to use? **License**: Which open-source protocol to use? Note that the short line represents partial support.

| Framework | Modularized | Parallel | Decoupling | Backend | License |
|-----------|:-----------:|:--------:|:----------:|:-------:|:-------:|
| Baselines | ✓ | ✗ | - | TensorFlow | MIT |
| SB3 | ✓ | ✗ | - | PyTorch | MIT |
| CleanRL | - | ✗ | ✗ | PyTorch | MIT |
| Ray/rllib | ✓ | ✓ | - | TF/PyTorch | Apache-2.0 |
| rlpyt | ✓ | ✓ | ✗ | PyTorch | MIT |
| Tianshou | ✓ | ✓ | - | PyTorch | MIT |
| ElegantRL | ✓ | ✓ | - | PyTorch | Apache-2.0 |
| SpinningUp | ✗ | ✗ | ✗ | PyTorch | MIT |
| ACME | ✗ | ✓ | ✗ | TF/JAX | Apache-2.0 |
| RLLTE | ✓ | ✓ | ✓ | PyTorch | MIT |

## 4  RELATED WORK

We compare RLLTE with eleven representative open-source RL projects, namely Baselines (Dhariwal et al., 2017), SB3 (Raffin et al., 2021), CleanRL (Huang et al., 2022), Ray/rllib (Liang et al.,

2018), and rlpyt (Stooke & Abbeel, 2019), Tianshou (Weng et al., 2022a), ElegantRL (Liu et al., 2021), SpinningUp (Achiam, 2018), and ACME (Hoffman et al., 2020), respectively. The following comparison is conducted from three aspects: architecture, functionality, and engineering quality. This project references some other open-source projects and adheres to their open-source protocols.

Table 5: Functionality comparison with existing projects. **Custom Env.**: Support custom environments? Since Gym (Brockman et al., 2016) is no longer maintained, it is critical to make the project adapt to Gymnasium (Towers et al., 2023). **Custom Module**: Support custom modules? **Data Aug.**: Support data augmentation techniques like intrinsic reward shaping and observation augmentation? **Data Hub**: Have a data hub to store benchmark data? **Deploy.**: Support model deployment? **Eval.**: Support model evaluation? **Multi-Device**: Support hardware acceleration of different computing devices (e.g., GPU and NPU)? Note that the short line represents partial support.

| Framework | Number of Algo. | Custom Env. | Custom Module | Data Aug. | Data Hub | Deploy. | Eval. | Multi-Device |
|---|---|---|---|---|---|---|---|---|
| Baselines | 9 | ✓(gym) | - | ✗ | - | ✗ | ✗ | ✗ |
| SB3 | 7 | ✓(gymnasium) | - | - | ✓ | ✗ | ✗ | ✗ |
| CleanRL | 9 | ✗ | ✓ | - | ✓ | ✗ | ✗ | ✗ |
| Ray/rllib | 16 | ✓(gym) | - | - | - | ✗ | ✗ | ✗ |
| rlpyt | 11 | ✗ | - | ✗ | - | ✗ | ✗ | ✗ |
| Tianshou | 20 | ✓(gymnasium) | ✗ | - | - | ✗ | ✗ | ✗ |
| ElegantRL | 9 | ✓(gym) | ✗ | ✗ | - | ✗ | ✗ | ✗ |
| SpinningUp | 6 | ✓(gym) | ✗ | ✗ | - | ✗ | ✗ | ✗ |
| ACME | 14 | ✓(dm_env) | ✗ | ✗ | - | ✗ | ✗ | ✗ |
| RLLTE | 13↗ | ✓(gymnasium) | ✓ | ✓ | ✓ | ✓ | ✓ | ✓ |

Table 6: Engineering quality comparison with existing projects. Note that the short line represents unknown.

| Framework | Documentation | Code Coverage | Type Hints | Last Update | Used by |
|---|---|---|---|---|---|
| Baselines | ✗ | ✗ | ✗ | 01/2020 | 508 |
| SB3 | ✓ | 96% | ✓ | 09/2023 | 3.3k |
| CleanRL | ✓ | - | ✗ | 09/2023 | 27 |
| Ray/rllib | ✓ | - | ✗ | 09/2023 | - |
| rlpyt | ✓ | 15% | ✗ | 09/2020 | - |
| Tianshou | ✓ | 91% | ✓ | 09/2023 | 169 |
| ElegantRL | ✓ | - | ✓ | 07/2023 | 256 |
| SpinningUp | ✓ | ✗ | ✗ | 02/2020 | - |
| ACME | ✓ | - | ✗ | 07/2023 | 149 |
| RLLTE | ✓ | 97% | ✓ | 09/2023 | 2↗ |

## 5 DISCUSSION

In this paper, we introduced a novel RL framework entitled RLLTE, which is a long-term evolution, extremely modular, and open-source project for advancing RL research and applications. With a rich and comprehensive ecosystem, RLLTE enables developers to accomplish task design, model training, evaluation, and deployment within one framework seamlessly, which is highly stimulative for both academia and industry. Moreover, RLLTE is an ultra-open framework where developers can freely use and try many built-in or custom modules, contributing to the research of decoupling and interpretability of RL. As a long-term evolution project, RLLTE will keep tracking the latest research progress and provide high-quality implementations to inspire more subsequent research.

In particular, there are some remaining issues that we intend to work on in the future. Firstly, RLLTE plans to add more algorithm prototypes to meet the task requirements of different scenarios, including multi-agent RL, inverse RL, imitation learning, and offline RL. Secondly, RLLTE will enhance

the functionality of the pre-training module, which includes more prosperous training methods and more efficient training processes, as well as providing downloadable model parameters. Thirdly, RLLTE will further explore the combination of RL and LLM, including using LLM to control the construction of RL applications and improving the performance of existing algorithms (e.g., reward function design and data generation). Finally, RLLTE will optimize the operational efficiency of modules at the hardware level to reduce the computational power threshold, promoting the goal of RL for everyone.

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

# A FUNCTION LIST

## A.1 XPLOIT: MODULES THAT FOCUS ON EXPLOITATION IN RL.

Table 7: `rllte.xploit.policy`: Policies for interaction and learning.

| Module | Type | Remark |
|---|---|---|
| `OnPolicySharedActorCritic` | On-policy | Actor-Critic networks with a shared encoder. |
| `OnPolicyDecoupledActorCritic` | On-policy | Actor-Critic networks with two separate encoders. |
| `OffPolicyDoubleQNetwork` | Off-policy | Double Q-network. |
| `OffPolicyDetActorDoubleCritic` | Off-policy | Deterministic actor network and double-critic network. |
| `OffPolicyDoubleActorDoubleCritic` | Off-policy | Double-actor network and double-critic network. |
| `OffPolicyStochActorDoubleCritic` | Off-policy | Stochastic actor network and double-critic network. |
| `DistributedActorLearner` | Distributed | Memory-shared actor and learner networks. |

Table 8: `rllte.xploit.encoder`: Neural nework-based encoders for processing observations. **Naming Rule**: Surname of the first author + Backbone + Encoder. **Target Task**: The testing tasks reported in their paper or potential tasks.

| Module | Input | Target Task |
|---|---|---|
| `EspeholtResidualEncoder` (Espeholt et al., 2018) | Images | Atari or Procgen games |
| `MnihCnnEncoder` (Mnih et al., 2013) | Images | Atari games |
| `TassaCnnEncoder` (Tassa et al., 2018) | Images | DMC Suite: pixel |
| `PathakCnnEncoder` (Pathak et al., 2017) | Images | Atari or MiniGrid games |
| `IdentityEncoder` | States | DMC Suite: state |
| `VanillaMlpEncoder` | States | DMC Suite: state |
| `RaffinCombinedEncoder` (Raffin et al., 2021) | Dict | Highway |

Table 9: `rllte.xploit.storage`: Experience storage and sampling.

| Module | Type |
|---|---|
| `VanillaRolloutStorage` | On-policy |
| `DictRolloutStorage` | On-policy |
| `VanillaReplayStorage` | Off-policy |
| `DictReplayStorage` | Off-policy |
| `NStepReplayStorage` (Sutton & Barto, 2018) | Off-policy |
| `PrioritizedReplayStorage` (Schaul et al., 2016) | Off-policy |
| `HerReplayStorage` (Andrychowicz et al., 2017) | Off-policy |
| `VanillaDistributedStorage` | Distributed |

## A.2 XPLORE: MODULES THAT FOCUS ON EXPLORATION IN RL.

Table 10: `rllte.xploit.augmentation`: PyTorch.nn-like modules for observation augmentation.

| Module | Input |
| --- | --- |
| `GaussianNoise` (Laskin et al., 2020) | States |
| `RandomAmplitudeScaling` (Laskin et al., 2020) | States |
| `GrayScale` (Laskin et al., 2020) | Images |
| `RandomColorJitter` (Laskin et al., 2020) | Images |
| `RandomConvolution` (Laskin et al., 2020) | Images |
| `RandomCrop` (Laskin et al., 2020) | Images |
| `RandomCutout` (Laskin et al., 2020) | Images |
| `RandomCutoutColor` (Laskin et al., 2020) | Images |
| `RandomFlip` (Laskin et al., 2020) | Images |
| `RandomRotate` (Laskin et al., 2020) | Images |
| `RandomShift` (Yarats et al., 2021) | Images |
| `RandomTranslate` (Laskin et al., 2020) | Images |

Table 11: `rllte.xploit.distribution`: Distributions for sampling actions. In RLLTE, the action noise is implemented via a distribution manner to realize unification.

| Module | Type |
| --- | --- |
| `NormalNoise` | Noise |
| `OrnsteinUhlenbeckNoise` | Noise |
| `TruncatedNormalNoise` | Noise |
| `Bernoulli` | Distribution |
| `Categorical` | Distribution |
| `MultiCategorical` | Distribution |
| `DiagonalGaussian` | Distribution |
| `SquashedNormal` | Distribution |

Table 12: `rllte.xploit.reward`: Intrinsic reward modules for enhancing exploration.

| Type | Modules |
| --- | --- |
| Count-based | `PseudoCounts` (Badia et al., 2020), `RND` (Burda et al., 2019) |
| Curiosity-driven | `ICM` (Pathak et al., 2017) (Pathak et al., 2017), `GIRM` (Yu et al., 2020), `RIDE` (Raileanu et al., 2020) |
| Memory-based | `NGU` (Badia et al., 2020) |
| Information theory-based | `RE3` (Seo et al., 2021), `RISE` (Yuan et al., 2022b), `REVD` (Yuan et al., 2022a) |

# B  Implemented RL Algorithms

Table 13: Implemented RL algorithms using RLLTE modules. **Dis., M.B., and M.D.**: Discrete, multi-binary, and multi-discrete action space. **M.P.**: Multi processing. **I.R.**: Support intrinsic reward shaping. **O.A.**: Support observation augmentation.

| Type | Algo. | Box | Dis. | M.B. | M.D. | M.P. | NPU | I.R. | O.A. |
|------|-------|-----|------|------|------|------|-----|------|------|
| On-Policy | A2C | ✓ | ✓ | ✓ | ✓ | ✓ | ✓ | ✓ | ✗ |
| On-Policy | PPO | ✓ | ✓ | ✓ | ✓ | ✓ | ✓ | ✓ | ✗ |
| On-Policy | DrAC | ✓ | ✓ | ✓ | ✓ | ✓ | ✓ | ✓ | ✓ |
| On-Policy | DAAC | ✓ | ✓ | ✓ | ✓ | ✓ | ✓ | ✓ | ✗ |
| On-Policy | DrDAAC | ✓ | ✓ | ✓ | ✓ | ✓ | ✓ | ✓ | ✓ |
| On-Policy | PPG | ✓ | ✓ | ✓ | ✗ | ✓ | ✓ | ✓ | ✓ |
| Off-Policy | DQN | ✓ | ✗ | ✗ | ✗ | ✓ | ✓ | ✓ | ✗ |
| Off-Policy | DDPG | ✓ | ✗ | ✗ | ✗ | ✓ | ✓ | ✓ | ✗ |
| Off-Policy | TD3 | ✓ | ✗ | ✗ | ✗ | ✓ | ✓ | ✓ | ✗ |
| Off-Policy | SAC | ✓ | ✗ | ✗ | ✗ | ✓ | ✓ | ✓ | ✗ |
| Off-Policy | SAC-Discrete | ✗ | ✓ | ✗ | ✗ | ✓ | ✓ | ✓ | ✗ |
| Off-Policy | DrQ-v2 | ✓ | ✗ | ✗ | ✗ | ✗ | ✓ | ✓ | ✓ |
| Distributed | IMPALA | ✓ | ✓ | ✗ | ✗ | ✓ | ✗ | ✗ | ✗ |

Full names and references of all algorithms:

- **A2C**: Advantage Actor-Critic (Mnih et al., 2016).
- **PPO**: Proximal Policy Optimization (Schulman et al., 2017).
- **DrAC**: Data-Regularized Actor-Critic (Raileanu et al., 2021).
- **DAAC**: Decoupled Advantage Actor-Critic (Raileanu & Fergus, 2021).
- **DrDAAC**: The combination of DrAC and DAAC.
- **PPG**: Phasic Policy Gradient (Cobbe et al., 2021).
- **DQN**: Deep Q-Network (Mnih et al., 2013).
- **DDPG**: Deep Deterministic Policy Gradient (Lillicrap et al., 2016).
- **TD3**: Twin Delayed DDPG (Fujimoto et al., 2018).
- **SAC**: Soft Actor-Critic (Haarnoja et al., 2018).
- **SAC-Discrete**: Soft Actor-Critic (Discrete) (Christodoulou, 2019).
- **DrQ-v2**: Data-Regularized Q-v2 (Yarats et al., 2021).
- **IMPALA**: Importance Weighted Actor-Learner Architecture (Espeholt et al., 2018).

## C  CODE EXAMPLES OF THE APPLICATION LAYER

### C.1  FAST ALGORITHM CONSTRUCTION

```python
from rllte.common.prototype import OnPolicyAgent
from rllte.xploit.encoder import MnihCnnEncoder
from rllte.xploit.policy import OnPolicySharedActorCritic
from rllte.xploit.storage import VanillaRolloutStorage
from rllte.xplore.distribution import Categorical

from torch import nn
import torch as th

class A2C(OnPolicyAgent):
    def __init__(self, env, tag, seed, device, num_steps) -> None:
        super().__init__(env=env, tag=tag, seed=seed, device=device, num_steps=num_steps)
        # create modules
        encoder = MnihCnnEncoder(observation_space=env.observation_space, feature_dim=512)
        policy = OnPolicySharedActorCritic(observation_space=env.observation_space,
                                           action_space=env.action_space,
                                           feature_dim=512,
                                           opt_class=th.optim.Adam,
                                           opt_kwargs=dict(lr=2.5e-4, eps=1e-5),
                                           init_fn="xavier_uniform"
                                           )
        storage = VanillaRolloutStorage(observation_space=env.observation_space,
                                        action_space=env.action_space,
                                        device=device,
                                        storage_size=self.num_steps,
                                        num_envs=self.num_envs,
                                        batch_size=256
                                        )
        dist = Categorical()
        # set all the modules
        self.set(encoder=encoder, policy=policy, storage=storage, distribution=dist)

    def update(self):
        for _ in range(4):
            for batch in self.storage.sample():
                # evaluate the sampled actions
                new_values, new_log_probs, entropy = \
                    self.policy.evaluate_actions(obs=batch.observations, actions=batch.actions)
                # policy loss part
                policy_loss = - (batch.adv_targ * new_log_probs).mean()
                # value loss part
                value_loss = 0.5 * (new_values.flatten() - batch.returns).pow(2).mean()
                # update
                self.policy.optimizers['opt'].zero_grad(set_to_none=True)
                (value_loss * 0.5 + policy_loss - entropy * 0.01).backward()
                nn.utils.clip_grad_norm_(self.policy.parameters(), 0.5)
                self.policy.optimizers['opt'].step()
```

Figure 6: Implement A2C algorithm with dozens of lines of code.

## C.2 MODULE REPLACEMENT

### C.2.1 USE BUILT-IN MODULES

```python
from rllte.agent import PPO
from rllte.env import make_atari_env

if __name__ == "__main__":
    # env setup
    device = "cuda:0"
    env = make_atari_env(device=device)
    eval_env = make_atari_env(device=device)
    # create agent
    agent = PPO(env=env,
                eval_env=eval_env,
                device=device,
                tag="ppo_atari")
    # start training
    agent.train(num_train_steps=5000)
```

```python
from rllte.agent import PPO
from rllte.env import make_atari_env
from rllte.xploit.encoder import EspeholtResidualEncoder

if __name__ == "__main__":
    # env setup
    device = "cuda:0"
    env = make_atari_env(device=device)
    eval_env = make_atari_env(device=device)
    # create agent
    feature_dim = 512
    agent = PPO(env=env,
                eval_env=eval_env,
                device=device,
                tag="ppo_atari",
                feature_dim=feature_dim)
    # create a new encoder
    encoder = EspeholtResidualEncoder(
        observation_space=env.observation_space,
        feature_dim=feature_dim)
    # set the new encoder
    agent.set(encoder=encoder)
    # start training
    agent.train(num_train_steps=5000)
```

Figure 7: **Left**: Train an PPO agent on the Atari games. **Right**: Replace the default encoder with `EspeholtResidualEncoder`.

### C.2.2 USE CUSTOM MODULES

```python
from rllte.agent import PPO
from rllte.env import make_atari_env
from rllte.common.prototype import BaseEncoder
from gymnasium.spaces import Space
from torch import nn
import torch as th

class CustomEncoder(BaseEncoder):
    """Custom encoder.

    Args:
        observation_space (Space): The observation space of environment.
        feature_dim (int): Number of features extracted.

    Returns:
        The new encoder instance.
    """
    def __init__(self, observation_space: Space, feature_dim: int = 0) -> None:
        super().__init__(observation_space, feature_dim)

        obs_shape = observation_space.shape
        assert len(obs_shape) == 3

        self.trunk = nn.Sequential(
            nn.Conv2d(obs_shape[0], 32, 3, stride=2), nn.ReLU(),
            nn.Conv2d(32, 32, 3, stride=2), nn.ReLU(),
            nn.Flatten(),
        )

        with th.no_grad():
            sample = th.ones(size=tuple(obs_shape)).float()
            n_flatten = self.trunk(sample.unsqueeze(0)).shape[1]

        self.trunk.extend([nn.Linear(n_flatten, feature_dim), nn.ReLU()])

    def forward(self, obs: th.Tensor) -> th.Tensor:
        h = self.trunk(obs / 255.0)

        return h.view(h.size()[0], -1)
```

Figure 8: Define a custom CNN-based encoder with RLLTE. This encoder can automatically compute the dimension of the extracted features of observations.

# D  CODE EXAMPLES OF THE TOOL LAYER

## D.1  ENVIRONMENT DESIGN

```python
from rllte.agent import DrQv2
from rllte.env.utils import make_rllte_env
import gymnasium as gym
import numpy as np

class CustomEnv(gym.Env):
    def __init__(self, total_length) -> None:
        super().__init__()
        self.observation_space = gym.spaces.Box(shape=(9, 84, 84),
            high=255.0, low=0., dtype=np.uint8)
        self.action_space = gym.spaces.Box(shape=(7,),
            high=1., low=-1., dtype=np.float32)
        self.total_length = total_length
        self.count = 0

    def step(self, action):
        obs = self.observation_space.sample()
        reward = np.random.rand()
        if self.count < self.total_length:
            terminated = truncated = False
        else:
            terminated = truncated = True
        info = {"discount": 0.99}
        self.count += 1

        return obs, reward, terminated, truncated, info

    def reset(self, seed=None, options=None):
        self.count = 0
        return self.observation_space.sample(), {"discount": 0.99}

if __name__ == "__main__":
    # env setup
    device = "cuda:0"
    env = make_rllte_env(env_id=CustomEnv,
                         device=device,
                         env_kwargs={'total_length': 499} # set env arguments
                         )
    eval_env = make_rllte_env(env_id=CustomEnv,
                             device=device,
                             env_kwargs={'total_length': 499} # set env arguments
                             )
    agent = DrQv2(env=env,
                eval_env=eval_env,
                device=device,
                tag="drqv2_dmc_pixel")
    agent.train(num_train_steps=5000)
```

Figure 9: Define a custom environment and perform training using DrQ-v2 agent.

## D.2 Model Evaluation

Firstly, Suppose we want to evaluate algorithm performance on the Procgen (Cobbe et al., 2020) benchmark. First, download the data from `rllte.hub`:

```python
# load packages
from rllte.evaluation import Performance, Comparison, min_max_normalize
from rllte.hub.datasets import Procgen, Atari
import numpy as np
# load scores
procgen = Procgen()
procgen_scores = procgen.load_scores()
print(procgen_scores.keys())
# get ppo-normalized scores
ppo_norm_scores = dict()
MIN_SCORES = np.zeros_like(procgen_scores['ppo'])
MAX_SCORES = np.mean(procgen_scores['ppo'], axis=0)
for algo in procgen_scores.keys():
    ppo_norm_scores[algo] = min_max_normalize(procgen_scores[algo],
                                              min_scores=MIN_SCORES,
                                              max_scores=MAX_SCORES)
```

Figure 10: Download benchmark data from `rllte.hub`.

```python
# initialize the performance evaluator
perf = Performance(scores=ppo_norm_scores['ppo'],
                   get_ci=True # get confidence intervals
                   )
# computes mean of sample mean scores per task
print(perf.aggregate_mean())
# computes median of sample mean scores per task
print(perf.aggregate_median())
# computes optimality gap across all runs and tasks
print(perf.aggregate_og())
# computes the interquartile mean across runs and tasks
print(perf.aggregate_iqm())
```

Figure 11: Performance evaluation of single algorithm.

```python
# initialize the performance comparer
comp = Comparison(scores_x=ppo_norm_scores['ppg'],
                  scores_y=ppo_norm_scores['ppo'],
                  get_ci=True)
# compute the overall probability of imporvement of algorithm `X` over `Y`.
print(comp.compute_poi())
```

Figure 12: Performance comparison of multiple algorithms.

```python
from rllte.evaluation import (plot_interval_estimates,
                              plot_probability_improvement,
                              plot_sample_efficiency_curve,
                              plot_performance_profile)

# 1. plot various performance metrics of algorithms with stratified confidence intervals
# construct a performance dict
aggregate_performance_dict = {
    "MEAN": {},
    "MEDIAN": {},
    "IQM": {},
    "OG": {}
}
for algo in ppo_norm_scores.keys():
    perf = Performance(scores=ppo_norm_scores[algo], get_ci=True)
    aggregate_performance_dict['MEAN'][algo] = perf.aggregate_mean()
    aggregate_performance_dict['MEDIAN'][algo] = perf.aggregate_median()
    aggregate_performance_dict['IQM'][algo] = perf.aggregate_iqm()
    aggregate_performance_dict['OG'][algo] = perf.aggregate_og()

fig, axes = plot_interval_estimates(aggregate_performance_dict,
                                    metric_names=['MEAN', 'MEDIAN', 'IQM', 'OG'],
                                    algorithms=['ppg', 'mixreg', 'ppo', 'idaac', 'plr', 'ucb-drac'],
                                    xlabel="PPO-Normalized Score")
fig.savefig('./plot_interval_estimates1.png', format='png', bbox_inches='tight')

# 2. plots probability of improvement with stratified confidence intervals.
# construct a comparison dict
pairs = [['idaac', 'ppg'], ['idaac', 'ucb-drac'], ['idaac', 'ppo'],
    ['ppg', 'ppo'], ['ucb-drac', 'plr'],
    ['plr', 'mixreg'], ['ucb-drac', 'mixreg'],  ['mixreg', 'ppo']]

probability_of_improvement_dict = {}
for pair in pairs:
    comp = Comparison(scores_x=ppo_norm_scores[pair[0]],
                      scores_y=ppo_norm_scores[pair[1]],
                      get_ci=True)
    probability_of_improvement_dict['_'.join(pair)] = comp.compute_poi()

fig, ax = plot_probability_improvement(poi_dict=probability_of_improvement_dict)
fig.savefig('./plot_probability_improvement.png', format='png', bbox_inches='tight')
```

Figure 13: Two examples of the visualization tool of `rllte.evaluation`.

### D.3 RLLTE HUB

```python
from rllte.hub.datasets import Procgen

procgen = Procgen()
# load final scores
# For each algorithm, this will return a `NdArray` of size (10 x 16)
# where `scores[n][m]` represent the score on run `n` of task `m`.
procgen_scores = procgen.load_scores()
print(procgen_scores['ppo'].shape)

# load learning curves
# this will return the learning curves by a Python `Dict` like:
# curves = {
#     "ppo": {
#         "train": {"bigfish": np.ndarray(shape=(Number of seeds, Number of points)), ...},
#         "eval": {"bigfish": np.ndarray(shape=(Number of seeds, Number of points)), ...},
#     },
#     "daac": {
#         "train": {"bigfish": np.ndarray(shape=(Number of seeds, Number of points)), ...},
#         "eval": {"bigfish": np.ndarray(shape=(Number of seeds, Number of points)), ...},
#     },
#     ...
# }
curves = procgen.load_curves()
print(curves['ppo']['train']['bigfish'].shape)
print(curves['ppo']['eval']['bigfish'].shape)
```

Figure 14: `rllte.hub.datasets` provides test scores and learning cures of various RL algorithms on different benchmarks.

```python
from rllte.hub.models import Procgen
from rllte.env import make_procgen_env
import torch as th
import numpy as np

if __name__ == "__main__":
    # env setup
    device = "cuda:0"
    env_id = "starpilot"
    seed = 1
    # download the model
    procgen = Procgen()
    agent = procgen.load_models(agent="ppo",
                                env_id=env_id,
                                seed=seed,
                                device=device)
    # create env
    env = make_procgen_env(env_id=env_id, device=device, num_envs=1, seed=seed)
    # evaluate the model
    obs, infos = env.reset(seed=seed)
    # run the model
    episode_rewards, episode_steps = list(), list()
    while len(episode_rewards) < 10:
        # the exported model outputs logits of the action distribution
        action = th.softmax(agent(obs), dim=1).argmax(dim=1)
        obs, rewards, terminateds, truncateds, infos = env.step(action)

        if "episode" in infos:
            indices = np.nonzero(infos["episode"]["l"])
            episode_rewards.extend(infos["episode"]["r"][indices].tolist())
            episode_steps.extend(infos["episode"]["l"][indices].tolist())

    print(f"mean episode reward: {np.mean(episode_rewards)}")
    print(f"mean episode length: {np.mean(episode_steps)}")

# output:
# mean episode reward: 30.0
# mean episode length: 296.1
```

Figure 15: `rllte.hub.models` provides trained models of various RL algorithms on different benchmarks.

