# OpenReview forum: "RLLTE: Long-Term Evolution Project of Reinforcement Learning"
_ICLR.cc/2024/Conference — Submitted to ICLR 2024_

### Official Review · Reviewer_N3a2 · 2023-10-30

**Soundness:** 2 fair
**Presentation:** 2 fair
**Contribution:** 1 poor
**Rating:** 3
**Confidence:** 4

**Summary:**

The work introduces RLLTE, a framework for RL research. It provides a modular approach to designing RL agents and additionally provides modules for evaluation, comparison and an interface to Huggingface to share benchmarking results.

**Strengths:**

* The idea of a modular RL framework is neat.
* The framework has implementations for many well-known algorithms.
* Using a Datahub is a great idea (as also shown by SB3 and CleanRL)

**Weaknesses:**

* The work does not present novel ideas
* The paper does not show any experiments showing the advantage of a modular framework. Neither are any experiments included that would show why one should prefer RLLTE over something like SB3.
* I don’t see how “RLLTE decouples RL algorithms from the exploitation-exploration perspective”. This claim seems wholly false.
* The work states multiple times that RLLTE is open-source or even ultra-open, but no link to code is given nor is a supplementary archive uploaded.
* The introduced modularity seems to simply introduce many more hyperparameters, thus offloading a lot of critical choices to a potential user. The work fails to discuss how this affects users or how hyperparameters should be treated in RLLTE.

**Questions:**

* Why is there no code available?
* How much better/worse is hyperparameter tuning with RLLTE?

---

### Official Review · Reviewer_ZtLn · 2023-10-30

**Soundness:** 3 good
**Presentation:** 2 fair
**Contribution:** 2 fair
**Rating:** 5
**Confidence:** 4

**Summary:**

The authors have created a modular, open-source framework for reinforcement learning. It includes a number of different modular layers each of which is highly flexible in how they are combined leading to easy to write code for efficient, easily-parallelisable model training and evaluation.

**Strengths:**

The framework is clearly presented, and there are good comparisons to other frameworks trying to do achieve similar goals. The evaluation modules in particular are well-thought out  and follow standards which are being pursued by the community.

**Weaknesses:**

While there are comparisons to other frameworks, it is not clear that all of the comparisons are up-to-date and this is a major issue. In particular SB3 does support parallel learning and hardware acceleration and while it doesn't natively support model deployment, it is explained in the documentation how to export models. With a user base of over 3k users, it clearly has a lot of momentum within the community, and so the authors would have to make a strong case that their system is genuinely superior to SB3. In addition, RL-Baselines3-zoo provides a training framework for SB3.

In addition to this, much of the language used within the paper feels like it is trying to oversell the framework. While I am sure that this is not the authors intensions, phrases such as "RLLTE has been thoughtfully designed", "Beyond delivering top-notch algorithm implementations", "RLLTE is the first RL framework to build a complete and luxuriant ecosystem". Such ideas should come through in the technical details, and not have to be sold using overly-confident language.

**Questions:**

It would be useful to see a side-by-side comparison between RLLTE and SB3 to see precisely why it is superior. If this can be shown unequivocally, then I believe that there is a lot more merit.

---

### Official Review · Reviewer_gfMY · 2023-11-01

**Soundness:** 3 good
**Presentation:** 3 good
**Contribution:** 3 good
**Rating:** 6
**Confidence:** 3

**Summary:**

The paper presents RLLTE, a long-term evolution, extremely modular, and open-source framework for reinforcement learning (RL) research and application. RLLTE decouples RL algorithms from the exploitation-exploration perspective and provides a large number of components to accelerate algorithm development and evolution. The framework serves as a toolkit for developing algorithms and is the first RL framework to build a complete ecosystem, including model training, evaluation, deployment, benchmark hub, and large language model (LLM)-empowered copilot. RLLTE is expected to set standards for RL engineering practice and be highly stimulative for industry and academia.

**Strengths:**

* Modular and customizable design, allowing for easy algorithm development and improvement.
* Long-term evolution plan, ensuring the framework stays up-to-date with the latest research.
* Comprehensive ecosystem, covering various aspects of RL research and application.
* Built-in support for data augmentation techniques, improving sample efficiency and generalization ability.
* Multi-hardware support, accommodating diverse computing hardware configurations.

**Weaknesses:**

* The paper does not provide a thorough comparison of RLLTE with other existing RL frameworks.
* The proposed LLM-empowered copilot is in its early stages and may not be as effective as expected.
* The paper does not discuss potential limitations or challenges in implementing the proposed framework.

**Questions:**

* How does RLLTE compare to other existing RL frameworks in terms of performance, modularity, and ease of use?
* What are the specific advantages of using RLLTE over other RL frameworks for different types of RL problems?
* How does the proposed LLM-empowered copilot improve the overall RL research and application process?
* Are there any potential limitations or challenges in implementing the proposed RLLTE framework that the authors have not discussed?

---

### Meta-Review · Area_Chair_wour · 2023-12-09

**Metareview:**

The paper presents RLLTE, a new open-source framework for reinforcement learning. While the reviewers appreciate the modularity of RLLTE, they have raised several questions concerning comparisons to other RL frameworks, a discussion of advantages compared to other frameworks, and details on the LLM-powered copilot. The authors have not responded to the questions from the reviewers.

**Justification For Why Not Higher Score:**

Comparisons to other commonly used RL frameworks are missing. Some claims on modularity seem misleading. No rebuttal was submitted.

**Justification For Why Not Lower Score:**

N/A

---

### Decision · Program_Chairs · 2024-01-16

Reject